# Transcriptomic profiling of susceptible and resistant flax seedlings after *Fusarium oxysporum* lini infection

Aleksandra Boba[1]*, Kamil Kostyn[2], Bartosz Kozak[2], Iwan Zalewski[1], Jan Szopa[1], Anna Kulma[1]

1 Faculty of Biotechnology, University of Wroclaw, Wroclaw, Poland, 2 Department of Genetics, Plant Breeding and Seed Production, Faculty of Life Sciences and Technology, Wroclaw University of Environmental and Plant Sciences, Wroclaw, Poland

* aleksandra.boba@uwr.edu.pl

## Abstract

In this study transcriptome was analyzed on two fibrous varieties of flax: the susceptible Regina and the resistant Nike. The experiment was carried out on 2-week-old seedlings, because in this phase of development flax is the most susceptible to infection. We analyzed the whole seedlings, which allowed us to recognize the systemic response of the plants to the infection. We decided to analyze two time points: 24h and 48h, because our goal was to learn the mechanisms activated in the initial stages of infection, these points were selected based on the previous analysis of chitinase gene expression, whose increase in time of *Fusarium oxysporum* lini infection has been repeatedly confirmed both in the case of flax and other plant species. The results show that although qualitatively the responses of the two varieties are similar, it is the degree of the response that plays the role in the differences of their resistance to *F. oxysporum*.

## Introduction

Flax (*Linum usitatissimum* L.) is a crop plant which provides valuable seeds, a source of oil and seedcakes, and straw–being the source of fiber and shives. In this regard, the plant appears a perfect, zero waste crop of numerous applications in different branches of industry. The growing interest in flax cultivation due to the new application of its products emerging thanks to the application of biotechnologies, is constantly being in danger of the plant's susceptibility to pathogenic microorganisms. One of the most serious threat to flax cultivation are fusarioses with *Fusarium oxysporum* f. sp. lini (Fol) being the most dangerous *Fusarium* species due to its high specificity to this plant [1]. It penetrates into the plant through the root system and then spreads using vascular bundles. The most characteristic symptoms of the disease can be observed in the phase of rapid growth of flax, then the tops of plants wilt, whole plants brown and die off. The development of the disease causes dieback of the seedlings and in the case of adult plants fusarium head blight. It is estimated that around 20% of flax cultivation loss is a result of fusariosis [2, 3].

**Data Availability Statement:** The filtered data (clean reads) of 24 samples (8 samples in 3 biological repeats) were stored in FASTQ format and were deposited in the NCBI Sequence Read

Archive SRP168336 (https://www.ncbi.nlm.nih.gov/sra/?term=PRJNA504749).

**Funding:** AK 2014/15/B/NZ9/00470 National Science Centre 2018/29/B/NZ9/00288 National Science Centre https://www.ncn.gov.pl The funders had no role in study design, data collection and analysis, decision to publish, or preparation of the manuscript.

**Competing interests:** The authors have declared that no competing interests exist.

During the eons of evolutionary race between phytopathogens and plants, unique and complex mechanisms of immune responses have been developed by the latter to cope with the continuously improving infection strategies of the plant invaders. While most of the pathogen attacks are overcome by non-host resistance, which relies on plant basal defense response incited by recognition of pathogen-associated molecular patterns (PAMPs) by the plant pattern recognition receptors (PRRs) localized on the plasma membrane to activate PAMP-triggered immunity (PTI) [4], some specifically adapted pathogens can overcome the first barrier by delivering effector proteins into plant cells to suppress the host basal defense. These host-specific attackers must face with secondary barrier, i.e. effector-triggered immunity (ETI) [5]. In ETI plant disease resistance genes (R-genes) encode specific receptors, which following recognition of an effector protein originating from the pathogen activate subsequent immune responses. Disease resistance controlled by the R gene(s) or qualitative resistance, usually delivers complete resistance to a specific pathogen or pathogen race [6]. On the other hand, quantitative host resistance ("horizontal" resistance) is often oligogenic. It is usually of lesser effect, but many studies on quantitative disease resistance have indicated its importance in crop disease improvement. Among various approaches undertaken to enhance plant's horizontal resistance to pathogens is production of different plant varieties. These varieties vary in both resistance to infection and plant's desired traits such as yield or nutritional value. The differences in the plants' response to stress give the opportunity to study the mechanisms behind their resistance. Aside other techniques of molecular genetics, like generation of transgenic plants or gene edition, which give the possibility to study usually one or few genes' role of plant's response, comparable analysis of two varieties of the same species allows to investigate its different aspects [7].

In recent years, two research teams have shown the analysis of the transcriptome in the interaction of flax with Fol. In 2016, the response of oil type of flax, CDC Bethune, was investigated in 4 time points (2 hpi, 4 hpi, 10 hpi, 18 hpi) [8]. In 2017, analysis of transcriptome of flax seedlings for 4 fibrous cultivars (2 resistant—Dakota and #3896) and two susceptible (AP5 and TOST) and their cross-breeds under Fol infection was conducted by a Russian research team. Only the root tips of seedlings after 48 hours from infection were analyzed and the analysis considered early local response in flax seedlings [9]. In this study transcriptome analysis was performed on two fibrous varieties of flax: the susceptible Regina and the resistant Nike [10, 11]. The experiment was carried out on 2-week-old seedlings, because in this phase of development flax is the most susceptible to infection. In contrast to the Russian team, which used only the root tips for analysis, thus examining the local response, we decided to analyze the whole seedlings, which allowed us to recognize the systemic response of the plants to the infection. We decided to analyze two time points: 24h and 48h, because our goal was to learn the mechanisms activated in the initial stages of infection, these points were selected based on the previous analysis of chitinase gene expression, whose increase in time of Fol infection has been repeatedly confirmed both in the case of flax and other plant species [12–14].

## Methods

### Plant material and experiment design

Flax seeds (*Linum usitatissimum* cv. Nike and cv. Regina), obtained from the Flax and Hemp Collection of the Institute of Natural Fibres in Poland, were grown on Murashige-Skoog (MS, Sigma-Aldrich) medium (with 1% sucrose and solidified with 0.9% agar) in Petri dishes and were left for 14 days in a phytotron chamber (16/8 h light/dark, light intensity ~100 µmol m$^{-2}$s$^{-1}$; 21˚C/16˚C day/night; and relative humidity 60%/70% day/night). *Fusarium oxysporum* linii (MYA-1201) were obtained from ATTC collection. *F. oxysporum* grow and

seedling treatments were conducted as described earlier with minor modifications [15]. 200 µl of *F. oxysporum* conidium suspension (1,76x10$^7$/ml) prepared as described by Di et al. [16] was spread on the petri dishes with PDA medium and cultivated for 2 days. Fourteen-day-old flax seedlings were moved (with the medium) onto *F. oxysporum*. The flax seedlings were collected after 24 h and 48 h and analyzed. The experiments were done in 3 biological repeats.

## RNA isolation and sequencing

Total RNA was extracted from the flax seedling tissue ground in liquid nitrogen using mir-Vana™ miRNA Isolation Kit (ThermoFisher) according to the producer's instruction. Genomic DNA was removed with DNase I (ThermoFisher). RNA quality was assessed using an Agilent 2100 Bioanalyzer (Agilent RNA 6000 Nano Kit). Generation of sequencing library requires a top-quality RNA to be isolated from the tissue of investigation. The RNA integrity (RIN) is of particular relevance as it positively correlates with mapped reads in RNAseq [17]. In this study, RNA samples with a RIN value > 7.5 were employed for RNAseq library construction, which meant that high-quality reads were obtained for subsequent studies (S1 Table). mRNA was isolated from the total RNA with oligo(dT) method. Then the mRNAs were fragmented under certain conditions and the first strand cDNA and the second strand cDNA were synthesized and joined with adapters. The cDNA fragments with suitable size were amplified with PCR and sequenced on Illumina HiSeq 2500 device.

## Bioinformatics workflow

Firstly, the low-quality reads (more than 20% of the bases qualities are lower than 30), reads with adaptors and reads with unknown bases (N bases more than 5%) were filtered using trimmomatic software [18] to get the clean reads. Basic statistics for both raw and clean reads are presented in S2 Table. The clean reads were aligned to reference genome [19] using Hisat2 software (v2.1; https://ccb.jhu.edu/software/hisat2/index.shtml) with the following parameters: "-q—phred64—dpad 0—gbar 99999999—mp 1,1—np 1—n-ceil L, 0,0.15—no-mixed—no-discordant -p 38 -k 10") [20]. The gene expression level was calculated using FeatureCounts software from Subread package (v1.6; http://subread.sourceforge.net/) [21]. Finally, DEGs (differential expressed genes) between samples were identified by DESeq2 [22]. The analysis pipeline is shown in S1 Fig.

## PCA

Raw count matrix generated by FeatureCounts were normalized with rlog function from DESeq2 package. Next those data were used in Principle Component Analysis (PCA). Calculation was performed in R Software with prcomp function and visualized with FactoMineR package [23] and ggplot2 packages [24].

## DEG detection and GO analysis

DESeq was used for differentially expressed genes (DEGs) with the following parameters: "Fold Change > = 2 and Adjusted P value < = 0.001". The Flax transcript from reference genome without assigned GO number were sought against genome database for the black cottonwood (*Populus trichocarpa*) (Torr. & Gray) using BLASTx algorithm. Obtained results were filtered using E value threshold (1e-40). The filtrated data were used to assign GO annotation to Flax transcript based on GO annotate to black cottonwood genes (S3 Table). DEGs were classified based on the GO annotation results and reference genome annotation. GO functional enrichment using goseq [25] package for R was also performed. False discovery rate

(FDR) for each p value was calculated. In general, the terms with FDR no larger than 0.01 were defined as significant enrichment. Hierarchical plots of GO terms were created using custom python script and goatools library [26] (S2 Fig).

## Results and discussion

### Data records

Transcriptomes of two flax cultivars–the resistant Nike and the susceptible Regina were sequenced in the seedlings exposed to *Fusarium oxysporum* infection for 24 h and 48 h in comparison to the non-treated control. The total read count ranged from 17.4 to 61.8 million for the analyzed samples of which about 96%-98% were clean reads.

The purpose of this study was to identify the differentially expressed genes under *F. oxysporum* infection (for changes of 2-fold or greater) of resistant (Nike) and susceptible (Regina) varieties of flax. PCA revealed good clustering of samples with a clear division between the treated and non-treated samples (Fig 1), in PC01 component. The differences between the resistant Nike and the susceptible Regina revealed by PCA analysis were similar for both infected and control plants. This indicates that as there are no big differences in responses to infections of Nike and Regina, changes in particular gene or gene group expression, that may be critical to plant's resistance should be sought out.

### Gene expression analysis in flax plants treated with *F. oxysporum*

Transcript levels in flax plants, both the susceptible and resistant cultivars, were evaluated in 24 hpi and 48 hpi in comparison to the control plants. Differential gene expression analysis was performed and the results can be found in S4 Table (Nike vs control at 24 hpi), S5 Table (Nike vs control at 48 hpi), S6 Table (Regina vs control at 24 hpi), S7 Table (Regina vs control at 48 hpi).

Analysis of the differentially expressed genes revealed differences in the number of up- and down-regulated genes between the resistant Nike and susceptible Regina cultivars and between different times of exposition to *F. oxysporum* (Fol). Higher number of genes was down-regulated in the Nike cultivar compared to Regina, while in Regina the numbers of down- and up-regulated were similar (Fig 2). Also, more genes in total were up/down regulated in the

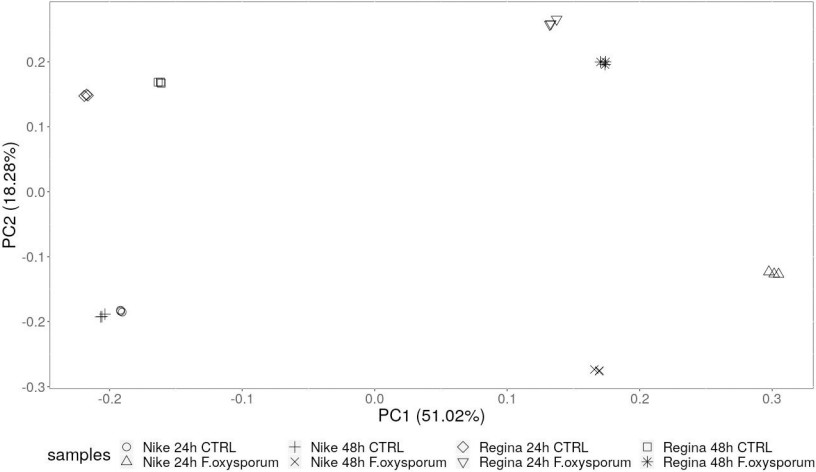

**Fig 1. Principle component analysis on transcriptome data from flax seedlings of the resistant and susceptible cultivars infected with *F. oxysporum*.**

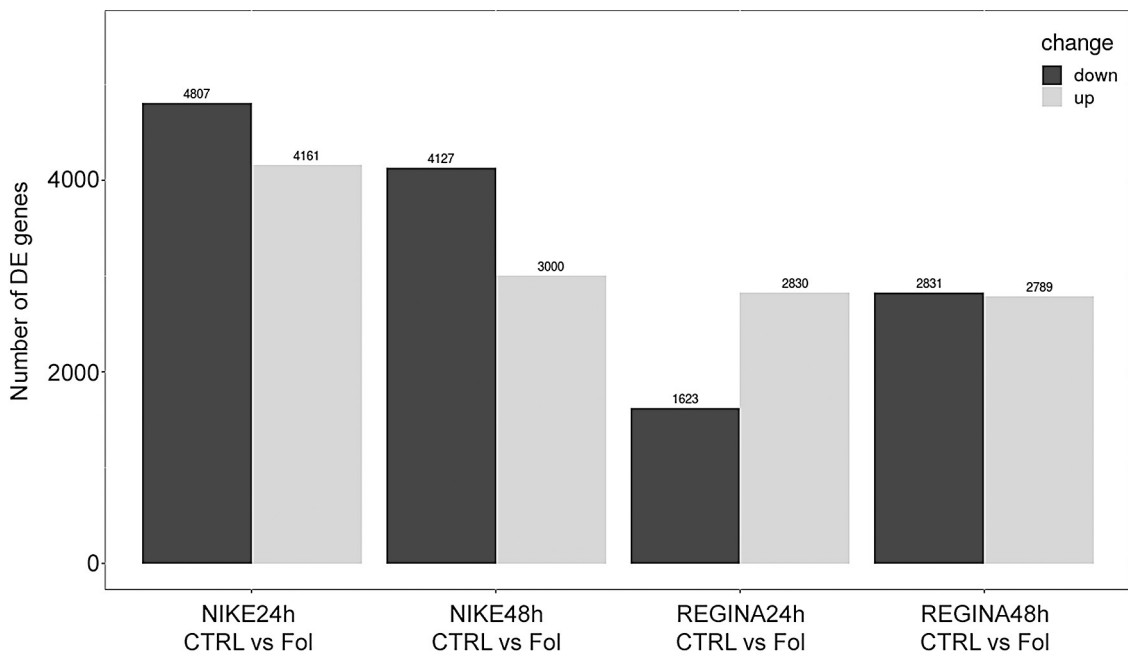

**Fig 2. Statistics of differentially expressed genes.**

resistant Nike cultivar. Environmental stress factors, such as pathogen infection, lead to dramatic reprogramming of transcription to favor stress responses over normal cellular functions. The bigger the changes in gene transcription the better the plant prepares to fend off the pathogen's attack [27].

## GO analysis of the resistant and susceptible cultivars

Gene ontology (GO) analysis was performed on differentially expressed genes (DEG) in Nike and Regina cultivars, at 24 hpi and 48 hpi. DEG number of GO terms (categories) that were statistically significantly overrepresented are provided in S8 Table. Hierarchical clustering of GO terms are presented in S2 Fig. The DEGs identified in the transcriptome analysis were classified in regard to the pathways they are involved in. The differences between Nike and Regina were not as clear as we expected. In fact, similar observation was made by Kroes et al. [11], where disease development in the resistant flax variety Hermes compared to Regina was similar. Among categories that counted the most up-regulated genes those involved in redox processes, signal transduction and specific binding to DNA were identified, both for Nike and Regina. This is not surprising as the genes are connected with early stages of plant's defense, like generation of ROS and signaling. However, in case of Nike a higher number of DEGs involved in these processes were observed. Upon infection, a plant recognizes specific molecules that after being registered trigger a sequence of signaling steps, leading to ion fluxes at the plasma membrane ($H^+/Ca^{2+}$ influxes, $K^+/Cl^-$ effluxes), ROS production, stimulation of protein kinase cascades, harnessing of specific transcription factors and consequently to activation of defense-associated gene expression [28]. We noted a higher number of differentially expressed genes involved in calcium signaling in the resistant Nike variety relatively to the susceptible Regina upon *Fusarium* infection (see in S9 Table and Table 1). A rapid increase in cytoplasmic free $Ca^{2+}$ levels is a common response to pathogen infection and $Ca^{2+}$ signal has been shown to be essential for the activation of defense responses, including oxidative burst

**Table 1. Number of DEGs that appear uniquely in Nike and Regina at 24 hpi and 48 hpi in selected groups of genes.**

| Gene group | NIKE_24 | REGINA_24 | NIKE_48 | REGINA_48 |
|---|---|---|---|---|
| calcium_signaling | 32 | 7 | 29 | 15 |
| chitinase | 15 | 1 | 10 | 8 |
| Et_TFs | 17 | 6 | 8 | 16 |
| ethylene_biosynthesis | 13 | 1 | 5 | 8 |
| glutaredoxin | 7 | 3 | 9 | 5 |
| glutathione_cycle | 15 | 8 | 15 | 9 |
| β-1,3-glucanase | 5 | 0 | 1 | 0 |
| JA_synthesis | 1 | 3 | 4 | 6 |
| JA_TFs | 2 | 0 | 0 | 1 |
| kinase | 262 | 30 | 166 | 100 |
| phosphatase | 95 | 7 | 54 | 16 |
| piSAgt | 2 | 0 | 0 | 0 |
| Tfs | 141 | 32 | 93 | 57 |
| thioredoxins | 9 | 3 | 9 | 5 |
| WRKY_TFs | 21 | 3 | 17 | 12 |

[29]. Two main enzymatic systems are thought to be responsible for the rapid increase of ROS in the cell, plasma membrane NADPH oxidases (respiratory burst oxidases–RBOs) and cell wall peroxidases [30]. Transcript numbers and levels of respective genes were comparable in Nike and Regina at both time points. However, generation of ROS in the oxidative burst occurs within few hours (or minutes in some cases) after the perception of pathogen [31], while the first time point analyzed in this study was 24 hpi, much later than peak transcription of these genes.

Following the oxidative burst is of activation ROS neutralizing machinery, thus we looked for the genes connected with maintaining redox homeostasis (thioredoxins and glutaredoxins, and those involved in glutathione cycle) (see in S9 Table and Table 1). Thioredoxins and glutaredoxins are groups of small proteins controlling the redox status in plant cell and play a significant role in plant's reaction to pathogen attack [32]. In *Arabidopsis*, expression of *AtTRX-h3* and *AtTRX-h5* can be induced by a pathogen and contributes to systemic acquired resistance. They increase reducing equivalents to generate the cellular reducing environment required for the conversion of NPR1 from a nonfunctional dimer or oligomer to a functional monomer. As a result, PR genes are expressed and SAR develops [33]. They also participate in the regulation of enzyme activity, and is involved in the regulation of transcription factor (TF) activity [34]. We observed a higher number of DEGs encoding both thioredoxins and glutaredoxins, as well as those connected with glutathione cycle in Nike than in Regina at both 24 hpi and 48 hpi.

Even if not under pathogen attack, a number of transcription activators involved in immune response are expressed in plant cells, however, they are kept inactive. When needed, they are activated thank to the action of different mechanisms, of which calcium signaling and redox status are considered to be the most important [27]. $Ca^{2+}$ fluxes appear to function both upstream and downstream of ROS production, and further, calcium-dependent phosphorylation events have also been proposed to occur both upstream and downstream of ROS production in response to pathogens [35].

Reversible phosphorylation of specific transcription factors, by a concerted action of protein kinases and phosphatases, may represent a mechanism for rapid and flexible regulation of selective gene expression. The number of kinases and phosphatases overexpressed after

infection was considerably higher in Nike than in Regina at both 24 hpi and 48 hpi (see in S9 Table and Table 1). Activation of TFs already present in the cell that leads to increased production of plant hormones, critical for the development of plant immune response, like salicylic acid, jasmonic acid and ethylene or abscisic acid [36, 37]. Generally, plant responses to biotrophic pathogens, which require live tissue to complete their life cycle, are regulated by the SA signaling pathway, whereas necrotrophic pathogens that degrade plant material are regulated by the ET and/or JA signaling pathways. However, mechanisms underlying resistance to hemi-biotrophic *F. oxysporum* are more complex and concern a network of phytohormone signaling [38]. Hormone dependent transcription factor synthesis occurs in order to facilitate the plant to cope with dynamics of the infection process. The number of differentially expressed transcription factor genes, both up- and down-regulated was higher in the infected Nike than Regina in relation to their controls at 24 hpi, but this was reversed at 48 hpi (see in S9 Table and Table 1). Moreover, the ratio of up- to down-regulated genes in Nike was 0.96 at 24 hpi and 0.44, at 48 hpi, while in Regina 2.6 and 0.9, respectively. Among the gamut of the transcription activators, WRKY transcription factors act in a complex defense response network as both positive and negative regulators [39]. The number of DEGs of the WRKY TFs was significantly higher in Nike than in Regina at both timepoints analyzed (see in S9 Table and Table 1). WRKY TFs are mainly induced by SA upon infection. However, no differences were found in the transcription of genes involved in its biosynthesis between the two varieties. SA is readily transformed into its conjugates, like volatile methyl-salicylate, which acts as signaling molecule or non-volatile glucosides, which act as their reservoir, though SA glucosides (SAGs) were also shown to be responsible for activating the rice defenses necessary for chemically induced disease resistance against blast fungus pathogens, and that SAGs possibly contribute to SAR by serving as a natural regulator in rice plants [40]. Overexpression of SA glucosyl transferase in *Arabidopsis* led to contradictory results, since the levels of free SA and SAG (as well as the glucose ester of SA) decreased rather than increased [41]. However, since SAG is considered as a transporting form of this hormone, its higher level is generally connected with swift rate of activation of the defense response throughout the plant [42]. Differentially expressed pathogen-inducible salicylic acid glucosyltransferase gene number was higher in Nike vs Regina (see in S9 Table and Table 1).

Differentially expressed ethylene-responsive TF gene transcript numbers were comparable in Nike and Regina at 24 hpi, however, at 48 hpi, it was Regina that was characterized by its higher number, implying that the ethylene-driven response acts longer in this variety. This correlated with the number and expression levels of genes involved in ethylene synthesis (ACS and ACO). Similarly, differentially expressed jasmonate-dependent TF gene numbers were also similar for both varieties at 24 hpi and 48 hpi. However, the level of transcription of these genes in Regina was on a significantly higher level. Also, expression of jasmonate O-methyl-transferase gene was at higher level in Regina, bot at 24 hpi and 48 hpi. Also, DEGs involved in jasmonic acid synthesis (lipoxygenase, allene oxide cyclase, 12-oxophytodienoic acid reductase), were slightly, but elevated more in Regina than in Nike after infection (see in S9 Table and Table 1).

Studying differences in expression patterns between the resistant Nike variety and susceptible Regina variety, we could not have omitted the pathogenesis-related (PR) genes, among which chitinases and β-1,3-glucanases play a significant role. Their expression is under control of various phytohormones, which is species- or organ-specific [43]. Numbers of differentially expressed chitinase and β-1,3-glucanase genes were higher in Nike compared to Regina variety at 24 hpi, while they were similar at 48 hpi (see in S9 Table and Table 1). It was previously shown that β-1,3-glucanase as well as chitinase are essential for flax resistance to *Fusarium* [37]

and transgenic flax plants overexpressing the β-1,3-glucanase gene showed lower susceptibility to this pathogen [44].

Up- and down-regulation of genes within the same GO category observed for a variety (Nike or Regina) may result from the activation of alternative routes within a pathway or even redirections to other pathways in response to the infection. Such phenomena might have appeared during the evolution of plants' sedentary mode of life, which requires high flexibility of their metabolism in response to biological stimuli. Environmental stress may not only alter the metabolic activity, but often reroutes biosynthetic pathways. For example, it is well known that alternative respiratory pathway plays an important role in plant thermogenesis, fruit ripening and responses to environmental stresses [45, 46]. Moreover, it is often that several gene isoforms exist in plant genome and they may be under control of differentially induced promotors. For instance, cinnamyl alcohol dehydrogenase gene isoforms, involved in lignin biosynthesis, were differentially expressed under *F. oxysporum* infection in flax [47]. Similarly, expression pattern of gene isoforms of cellulase synthase and cellulase, connected with cell wall remodeling, was changed in flax after *F. oxysporum* infection [2]. In another example, isoforms of genes involved in phenylpropanoid biosynthesis pathway were shown to be differentially expressed upon *F. oxysporum* infection of flax [15]. Transcript levels of genes may be also altered by the very pathogen, for instance, soybean pathogen caused alternative splicing of pre-mRNAs from 401 soybean genes, including defense-related genes [48].

## Conclusion

Plant response to infection, especially at its early stages can be perceived as a continuous arms race between the plant and microorganism, where every response of the plant meets a counter-response of the pathogen and *vice versa*. In such a struggle better preparation of a plant increases the chances of its successful overcoming the infection. In the case of the two varieties of flax studied in our research, this better preparation is connected with a greater flexibility of the transcriptome, which translates to a higher number of activated and repressed genes. A more determined transcriptomal response of the Nike cultivar, which is connected to a more diversified enzyme homolog pool and/or activation of alternative pathways, leads to its quicker and more effective response.

## Supporting information

**S1 Fig. Bioinformatics workflow.**
(TIF)

**S2 Fig. Hierarchical plots of GO terms.**
(ZIP)

**S1 Table. Sample test results.**
(DOCX)

**S2 Table. Basic statistics for raw and clean reads.**
(XLSX)

**S3 Table. GO annotation to Flax transcript.**
(XLSX)

**S4 Table. Nike treated with pathogen vs Nike non-treated (control) at 24 hpi.**
(XLSX)

**S5 Table. Nike treated with pathogen vs Nike non-treated (control) at 48 hpi.**
(XLSX)

**S6 Table. Regina treated with pathogen vs Regina non-treated (control) at 24 hpi.**
(XLSX)

**S7 Table. Regina treated with pathogen vs Regina non-treated (control) at 48 hpi.**
(XLSX)

**S8 Table. DEG number of GO terms (categories).**
(XLSX)

**S9 Table.**
(XLSX)

## Author Contributions

**Conceptualization:** Anna Kulma.

**Data curation:** Bartosz Kozak.

**Formal analysis:** Kamil Kostyn, Bartosz Kozak, Anna Kulma.

**Investigation:** Aleksandra Boba, Iwan Zalewski.

**Project administration:** Anna Kulma.

**Supervision:** Jan Szopa.

**Writing – original draft:** Aleksandra Boba, Kamil Kostyn, Iwan Zalewski.

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
