## [Decision Letter · Decision Letter 0]

3 Nov 2020

PONE-D-20-23624

Transcriptomic profiling of susceptible and resistant flax seedlings after Fusarium oxysporum lini infection.

PLOS ONE

Dear Dr. Boba,

Thank you for submitting your manuscript to PLOS ONE. After careful consideration, we feel that it has merit but does not fully meet PLOS ONE’s publication criteria as it currently stands. Therefore, we invite you to submit a revised version of the manuscript that addresses all the points raised during the review process.

Your manuscript has been evaluated by two expert reviewers, who have suggested a number of changes that should be relatively easy to make and will bring some important information to the main body of the manuscript. Please note that all results published in PLOS ONE must be supported by the data presented, hence the importance of one of the problems pointed out by Reviewer #1 ("lack of figures or tables within the main manuscript to support the narrative in the Results & Discussion section"), which I find particularly important.

We look forward to receiving your revised manuscript.

Kind regards,

Hector Candela, Ph.D.

Academic Editor

PLOS ONE

Journal Requirements:

Reviewers' comments:

Reviewer's Responses to Questions

**Comments to the Author**

1. Is the manuscript technically sound, and do the data support the conclusions?

Reviewer #1: Partly

Reviewer #2: Yes

2. Has the statistical analysis been performed appropriately and rigorously? 

Reviewer #1: I Don't Know

Reviewer #2: Yes

3. Have the authors made all data underlying the findings in their manuscript fully available?

Reviewer #1: Yes

Reviewer #2: Yes

4. Is the manuscript presented in an intelligible fashion and written in standard English?

Reviewer #1: Yes

Reviewer #2: Yes

5. Review Comments to the Author

Reviewer #1: I like this manuscript, and the fact that the authors have replicated the experiment carefully and deposited the raw data in NCBI SRA. I also believe that transcriptome reports are valuable, and need not be too long or detailed; so I think this report is sufficiently concise. However, I was surprised by the lack of figures or tables within the main manuscript to support the narrative in the Results & Discussion section. The material from Supplemental Table 7 (GO enrichment) should be moved into the main body of the manuscript, as well as figure or table with specific gene expression values or ratios for the genes mentioned in the Discussion (with statistical significance), and perhaps an original summary diagram could be added to capture other information described in the Results & Discussion section but not presented elsewhere, e.g. these are just some of the statements that should be supported by citations to figures or tables:

We noted a higher number of genes involved in calcium signaling in the resistant Nike variety relatively to the susceptible Regina upon Fusarium infection.

The number of kinases and phosphatases overexpressed after infection was considerably higher in Nike than in Regina at both 24 hpi and 48 hpi.

The number of differentially expressed transcription factor genes, both up- and down-regulated was higher in the infected Nike than Regina in relation to their controls at 24 hpi, but this was reversed at 48 hpi. Moreover, the ratio of up- to down-regulated genes in Nike was 0.96 at 24 hpi and 0.44, at 48 hpi, while in Regina 2.6 and 0.9, respectively.

The number of DEGs of the WRKY TFs was significantly higher in Nike than in Regina at both timepoints analyzed. WRKY TFs are mainly induced by SA upon infection. However, no differences were found in the transcription of genes involved in its biosynthesis between the two varieties.

Differentially expressed pathogen-inducible salicylic acid glucosyltransferase gene number was higher in Nike vs Regina, at both time points.

Differentially expressed ethylene-responsive TF gene transcript numbers were comparable in Nike and Regina at 24 hpi, however, at 48 hpi, it was Regina that was characterized by its higher number, implying that the ethylene-driven response acts longer in this variety.

Similarly, differentially expressed jasmonate-dependent TF gene numbers were also similar for both varieties at 24 hpi and 48 hpi. However, the level of transcription of these genes in Regina was on a significantly higher level.

Numbers of differentially expressed chitinase and β-1,3glucanase genes were higher in Nike compared to Regina variety at 24 hpi, while they were similar at 48 hpi.

etc.

==

Line 204 starts "thus we looked for the genes connected with maintaining redox homeostasis (thioredoxins and glutaredoxins, and those involved in glutathione cycle)." but after talking about expression of these genes in other systems, there is no mention of the results of the current experiment.

Reviewer #2: In general terms, this is a technically correct work and both the public results provided and the discussion can fuel future plant-pathogen studies. However, I consider appropriate to request the authors to include some changes:

[Line 27-60] : most of the Introduction section is scarcely referenced. More references should be included (if there is) in order to support the information exposed.

[Line 99-101] : software used for read cleaning should be specified, as well as average reads per sample and nucleotides long. Note that Hisat2 is an alignment program, not an assembler software; so, "The clean reads were "aligned""... instead of "assembled".

[Line 110] : the authors should consider removing Figure 1 or move it to Supplementary files, since the Bioinformatics workflow is the standard pipeline for transcriptomics analysis and there is no customized steps.

[Line 112-124] : DEG Detection and Analysis and GO Annotation sections should be merged or, at least, put together all the ontology analysis to make easier an overall understanding.

[Line 127-142] : RNA RIN parameters should be included within RNA isolation and sequencing section. In the same way, within Quality validation subsection, move only per base quality score statistics to the beginning of Bioinformatics workflow section.

[Line 145] : PCA methods should me moved to its corresponding section and just present and discuss the results.

[Line 174] : typographical error; "such like"  "such as".

[Line 180] : In this section, a graphical representation of overrepresented GO terms should be included in the main text using specific tools for annotation visualization and functional analysis (AgriGO, Blast2GO...).

6. PLOS authors have the option to publish the peer review history of their article (what does this mean?). If published, this will include your full peer review and any attached files.

Reviewer #1: No

Reviewer #2: No

---

## [Author Response · Author response to Decision Letter 0]

24 Nov 2020

Reviewer #1: I like this manuscript, and the fact that the authors have replicated the experiment carefully and deposited the raw data in NCBI SRA. I also believe that transcriptome reports are valuable, and need not be too long or detailed; so I think this report is sufficiently concise. However, I was surprised by the lack of figures or tables within the main manuscript to support the narrative in the Results & Discussion section. The material from Supplemental Table 7 (GO enrichment) should be moved into the main body of the manuscript, as well as figure or table with specific gene expression values or ratios for the genes mentioned in the Discussion (with statistical significance), and perhaps an original summary diagram could be added to capture other information described in the Results & Discussion section but not presented elsewhere, e.g. these are just some of the statements that should be supported by citations to figures or tables:

We noted a higher number of genes involved in calcium signaling in the resistant Nike variety relatively to the susceptible Regina upon Fusarium infection.

The number of kinases and phosphatases overexpressed after infection was considerably higher in Nike than in Regina at both 24 hpi and 48 hpi.

The number of differentially expressed transcription factor genes, both up- and down-regulated was higher in the infected Nike than Regina in relation to their controls at 24 hpi, but this was reversed at 48 hpi. Moreover, the ratio of up- to down-regulated genes in Nike was 0.96 at 24 hpi and 0.44, at 48 hpi, while in Regina 2.6 and 0.9, respectively.

The number of DEGs of the WRKY TFs was significantly higher in Nike than in Regina at both timepoints analyzed. WRKY TFs are mainly induced by SA upon infection. However, no differences were found in the transcription of genes involved in its biosynthesis between the two varieties.

Differentially expressed pathogen-inducible salicylic acid glucosyltransferase gene number was higher in Nike vs Regina, at both time points.

Differentially expressed ethylene-responsive TF gene transcript numbers were comparable in Nike and Regina at 24 hpi, however, at 48 hpi, it was Regina that was characterized by its higher number, implying that the ethylene-driven response acts longer in this variety.

Similarly, differentially expressed jasmonate-dependent TF gene numbers were also similar for both varieties at 24 hpi and 48 hpi. However, the level of transcription of these genes in Regina was on a significantly higher level.

Numbers of differentially expressed chitinase and β-1,3glucanase genes were higher in Nike compared to Regina variety at 24 hpi, while they were similar at 48 hpi.

etc.

> Supplementary Table S7 (now renamed to Supplementary Table S8) is quite extensive (note several tabs in the excel file) and moving it to the main body of the text will extensively enlarge the volume of the manuscript. We would like to propose keeping it in the supplementary material. However, if the Reviewer insists on moving it to the text, we will gladly fulfill the request.

Similarly, a Table with changes in gene expression ratio for the genes mentioned in the Discussion will be placed in the supplementary material (Supplementary Table S9) due to its large volume (we considered depicting the results as heatmaps, but as there are many genes in the groups mentioned in the Discussion, the Figure would be also too big, and in our opinion illegible). Additionally, we propose to place a table (Table 1) with the numbers of genes with significant changes expression within each group described in the Discussion unique for NIKE and REGINA varieties.

==

Line 204 starts "thus we looked for the genes connected with maintaining redox homeostasis (thioredoxins and glutaredoxins, and those involved in glutathione cycle)." but after talking about expression of these genes in other systems, there is no mention of the results of the current experiment.

> We complemented this part with appropriate text.

Reviewer #2: In general terms, this is a technically correct work and both the public results provided and the discussion can fuel future plant-pathogen studies. However, I consider appropriate to request the authors to include some changes:

[Line 27-60] : most of the Introduction section is scarcely referenced. More references should be included (if there is) in order to support the information exposed.

> We added more references to the Introduction section.

[Line 99-101] : software used for read cleaning should be specified, as well as average reads per sample and nucleotides long. Note that Hisat2 is an alignment program, not an assembler software; so, "The clean reads were "aligned""... instead of "assembled".

> We specified the software used for read cleaning and added the missing information in Supplementary Table S2 with basic statistics for raw and clean reads. We corrected “assembled” to “aligned”.

[Line 110] : the authors should consider removing Figure 1 or move it to Supplementary files, since the Bioinformatics workflow is the standard pipeline for transcriptomics analysis and there is no customized steps.

> We moved the Figure 1 to supplementary files.

[Line 112-124] : DEG Detection and Analysis and GO Annotation sections should be merged or, at least, put together all the ontology analysis to make easier an overall understanding.

> We combined the two subsections as suggested.

[Line 127-142] : RNA RIN parameters should be included within RNA isolation and sequencing section. In the same way, within Quality validation subsection, move only per base quality score statistics to the beginning of Bioinformatics workflow section.

> The RIN parameters were moved to ‘RNA isolation and sequencing’ subsection. The average per base quality score was placed in the Supplementary Table S2.

[Line 145] : PCA methods should me moved to its corresponding section and just present and discuss the results.

> PCA method is now described in the Methods section.

[Line 174] : typographical error; "such like"  "such as".

> We corrected it.

[Line 180] : In this section, a graphical representation of overrepresented GO terms should be included in the main text using specific tools for annotation visualization and functional analysis (AgriGO, Blast2GO...).

> A graphical representation of overexpressed GO terms (hierarchical plots) was created using custom python script and goatools library. However, since the figures are quite extensive, we propose to place them in the supplementary material (Supplementary Fig. S2). However, if the Reviewer wishes us to insert them in the main text body, we will gladly do it.

---

## [Decision Letter · Decision Letter 1]

13 Jan 2021

Transcriptomic profiling of susceptible and resistant flax seedlings after Fusarium oxysporum lini infection.

PONE-D-20-23624R1

Dear Dr. Boba,

We’re pleased to inform you that your manuscript has been judged scientifically suitable for publication and will be formally accepted for publication once it meets all outstanding technical requirements.

Kind regards,

Hector Candela, Ph.D.

Academic Editor

PLOS ONE

Additional Editor Comments (optional):

Reviewers' comments:

Reviewer's Responses to Questions

**Comments to the Author**

1. If the authors have adequately addressed your comments raised in a previous round of review and you feel that this manuscript is now acceptable for publication, you may indicate that here to bypass the “Comments to the Author” section, enter your conflict of interest statement in the “Confidential to Editor” section, and submit your "Accept" recommendation.

Reviewer #1: All comments have been addressed

Reviewer #2: (No Response)

2. Is the manuscript technically sound, and do the data support the conclusions?

Reviewer #1: (No Response)

Reviewer #2: Yes

3. Has the statistical analysis been performed appropriately and rigorously? 

Reviewer #1: (No Response)

Reviewer #2: Yes

4. Have the authors made all data underlying the findings in their manuscript fully available?

Reviewer #1: (No Response)

Reviewer #2: Yes

5. Is the manuscript presented in an intelligible fashion and written in standard English?

Reviewer #1: (No Response)

Reviewer #2: Yes

6. Review Comments to the Author

Reviewer #1: (No Response)

Reviewer #2: As I suggested in the first round of review, the authors took into account and made the changes proposed. However, I still highly recommend to incorporate the following modifications regarding to Figures included in the main text body:

- PCoA Fig. 1. should be moved to Supplementary Files, since it is not providing value-added visual information beyond the data described in the main text, as a quality control figure itself, and so, it is not of paramount importance so as to facilitate a better understanding.

- Contrary to the above-mentioned PCoA Figure, I strongly encourage the authors to make a more synthetic figure that put together all the gene ontology data. As an alternative, I suggest using a circular way of representation, such as the freely accesible CirGO (https://doi.org/10.1186/s12859-019-2671-2).

7. PLOS authors have the option to publish the peer review history of their article (what does this mean?). If published, this will include your full peer review and any attached files.

Reviewer #1: No

Reviewer #2: No

---

## [Editor Report · Acceptance letter]

15 Jan 2021

PONE-D-20-23624R1 

Transcriptomic profiling of susceptible and resistant flax seedlings after *Fusarium oxysporum* lini infection. 

Dear Dr. Boba:

I'm pleased to inform you that your manuscript has been deemed suitable for publication in PLOS ONE. Congratulations! Your manuscript is now with our production department. 

Kind regards, 

on behalf of

Dr. Hector Candela 

Academic Editor

PLOS ONE